# Spatiotemporal Patterns of Land-Use Changes in Lithuania

**Daiva Juknelienė [1,](https://)***, **Vaiva Kazanavičiūtė [2]**, **Jolanta Valčiukienė [1]**, **Virginija Atkocevičienė [1]** and **Gintautas Mozgeris [1]**

1   Agriculture Academy, Vytautas Magnus University, Studentų Str. 11, Akademija, 53361 Kaunas, Lithuania; jolanta.valciukiene@vdu.lt (J.V.); virginija.atkoceviciene@vdu.lt (V.A.); gintautas.mozgeris@vdu.lt (G.M.)
2   Department of Environmental Sciences, Faculty of Natural Sciences, Vytautas Magnus University, Universiteto Str. 10-314, Akademija, 53361 Kaunas, Lithuania; vaiva.kazanaviciute@stud.vdu.lt
*   Correspondence: daiva.jukneliene@vdu.lt

**Abstract:** The spatially explicit assessment of land use and land-use change patterns can identify critical areas and provide insights to improve land management policies and associated decisions. This study mapped the land uses and land-use changes in Lithuanian municipalities since 1971. Additionally, an analysis was conducted of three shorter periods, corresponding to major national land-use policy epochs. Data on land uses, available from the Lithuanian National Forest Inventory (NFI) and collected on an annual basis with the primary objective of conducting greenhouse gas (GHG) accounting and reporting for the land use, land-use change, and forestry (LULUCF) sectors, were explored. The overall trend in Lithuania during the last five decades has been an increase in the area of forest and built-up land and decrease in the area of producing land, meadow/pasture, wetlands, and other land uses. Nevertheless, the development trends for the proportions of producing land and meadow/pasture changed trajectories several times, and the breakpoints were linked with important dates in Lithuanian history and associated with the reorganization of land management and land-use relations. Global Moran's *I* statistic and Anselin Local Moran's *I* were used to check for global and local patterns in the distribution of land use in Lithuanian municipalities. The proportions of producing land and pasture/meadow remained spatially autocorrelated during the whole period analysed. Local spatial clusters and outliers were identified for all land-use types used in GHG inventories in the LULUCF sector at all the time points analysed. Ordinary least squares (OLS) regression was used to explain the land-use change trends during several historical periods due to differing land management policies, utilizing data from freely available databases as the regressors. The percentage of variance explained by the models ranged from 37 to 65, depending on the land-use type and the period in question.

**Keywords:** land use; land-use change; forests; producing land; grassland; spatial autocorrelation; regression

## 1. Introduction

The monitoring of land-use changes is a key way to understand and assess the dynamic processes in landscapes under different time and spatial scales. A fast-growing human population, the exhaustive use of resources, and increasing environmental concerns have made land-use change monitoring an important topic on the international research agenda [1,2]. The interaction between human activity and land-use changes is an increasing focus of researchers [3,4] due to their impacts on the climate [5], ecosystems [6], water resources [7], soil quality [8], and socioeconomic systems [9]. Land-use changes due to biophysical factors and human activities are accelerating in different regions of the world [10–12]. Even though the issues related to land-use changes are global and cause severe problems in many countries, the change patterns are dependent on local conditions due to numerous factors, such as policies, management, economics, culture, human behaviour, and the environment [13–17]. Thus, it is extremely important to understand the

processes shaping land-use changes at different scales, ranging from regional to global. Such knowledge is of critical importance to build the policies and management plans needed to understand and improve the land-use change trends [12,17–20].

Land-use change in Lithuania has always been dynamic. The radical political, economic, and social developments that took place in the country over the last half century undoubtedly had impacts on the land use. Official statistics indicate [21] that 45.6% of the country's area is covered by arable land, 33.5% by forest, 6.21% by meadows and natural pastures, 5.2% by wetland, 5.4% by built-up land, and 4.09% by other. The area proportions of all land-use types, except for agricultural land, have changed relatively steadily during the last five decades; however, the trends of producing land and grassland development changed their trajectories around 1990 and again in about 2005 [22]. The demand for up-to-date information on land cover and land-use changes is increasing due to rapid landscape development as a result of fast processes in the agricultural sector, the growth of urban areas, and the depopulation of some regions, followed by renaturalization [23]. To implement the European Landscape Convention (2020) [24], the Lithuanian authorities (the Environmental Protection Agency of the Ministry of Environment) conduct regular monitoring of landscape changes. Such monitoring delivers facts on landscape development peculiarities and the factors behind the trends, which are needed to predict potential future opportunities and risks [25]. Nevertheless, the data collected and methods of analyses differ from region to region. Scientific research in this area, to the best of our knowledge, has always been sparse. The changes of land cover structure were assessed on 100 test sites (totalling 2.5 km$^2$) in 1976–1986, 2005–2006, and 2012–2013 by the Institute of Geology and Geography (2008; 2015). Often, CORINE information was mobilized to assess the historical development of land cover [26–29]. Information related to land use in Lithuania may also be available from several nationwide GIS databases, such as the Spatial dataset of georeference base cadastre (GRPK) or the Land Parcel Identification System (KZS) Database, which are maintained by state institutions and available for free from the Spatial Information Portal of Lithuania (geoportal.lt). Together with the information on declared land uses and agricultural parcels, this could make an excellent land-use dataset for scientific research; however, such data are only available from 2010 onward. Usually, only the most recent version of the data is freely available. Thus, the availability of suitable data could be another reason behind the limited research focus on land-use retrospection.

Land use and its changes are not only important for the development of the economy or the protection of the environment but are also recognized as having a significant impact on human-induced greenhouse gas (GHG) emissions [30,31]. Land use and its changes may result in GHG removal if certain active measures are applied, such as afforestation, reforestation, revegetation, etc. [32,33]. In order to estimate such emissions and removals, the land use, land-use change, and forestry (LULUCF) sector's GHG reporting was included under the requirements of UNFCCC reporting. Despite the sector's ability to capture GHG emissions from the atmosphere and sequestrate it in biomass or soil, the LULUCF sector was not included in the climate change mitigation target until 2021 [34]. Beginning in 2021, the LULUCF sector will play a role in the flexibility option to reach compliance with other sectors' GHG emission reduction target.

To meet its international climate change mitigation commitments and fulfil the obligation of reporting on GHG emissions and removals in the LULUCF sector, Lithuania introduced an original land-use monitoring system, which became an integral part of the National Forest Inventory (NFI), implemented by the State Forest Service [35,36]. The inventory uses a network of 16,349 systematically allocated sampling points. The land-use type and subtype were identified at each point following the Good Practice Guidance for Land Use, Land-Use Change and Forestry (IPCC 2003), also taking into consideration the requirements of the United Nations Framework Convention on Climate Change and the Kyoto protocol for each year starting in 1971. Past land uses at each point were identified using available historical maps, such as topographic maps, land management maps, orthophotos, or satellite images [37]. The information collected in the sampling plots was

used to prepare a land use and land-use change database, in addition to conventional forestry statistics, traditionally attributed to forest inventories. This information has been used in Lithuania to conduct greenhouse gas (GHG) accounting and reporting in the Land Use, Land-Use Change, and Forestry (LULUCF) sector since 2010. Usually, conventional land-use-data-based exercises are based on aggregated statistical information at the country level. Considerable spatial patterns of land-use distribution may be seen in a relatively small country such as Lithuania.

The fast progress of geographic information systems (GIS) during the last few decades provided researchers with powerful tools with which to conduct spatial analyses and modelling [38]. In Lithuania, there were few attempts to use GIS as a tool in land-use-related studies. For example, Kucas et al. [39] applied a multiscale analysis of forest fragmentation in Lithuania to demonstrate the technique with CORINE data. Lazdinis et al. [40] suggested an alternative—the average shortest distance to the closest forest—to forest cover percentage, better describing the spatial distribution of forested habitats for birds in an afforestation study. Jukneliene and Mozgeris [41] compared two GIS databases, representing the forest cover at a nominal scale of 1:10,000 and referring to two dates—1950 and 2013. The data were aggregated for the analyses up to the municipality level. The Global Moran's *I* statistic and Anselin Local Moran's *I* were used to identify global and local patterns in the distribution of forest cover characteristics in Lithuanian municipalities. The authors provided the reader with updated statistics on forest cover in Lithuania just after WWII and discussed the trends of forest cover dynamics during the second half of the 20th century. Recently, Manton et al. [42] used a local hotspot analysis to study peatlands in the Nemunas River basin. However, all these studies used wall-to-wall land-cover and land-use maps, referring to specific dates. The lack of continuously supplied information over time introduces some uncertainties in land-use change trajectories and, simultaneously, makes generalizing about land-use changes more challenging. A distinctive feature of the current study is that we analyse land-use data collected through sampling annually and covering the period since 1971. Another advantage of GIS is the opportunity to integrate for joint analysis data collected using different techniques, formats, time periods, and sometimes applications, but all sharing the same geographic location [22]. The availability of free multisource and multipurpose GIS data in the country has notably increased during the last decade since the implementation of the Spatial Information Portal of Lithuania [43]. All this potentially offers enhanced opportunities for a better understanding of the processes behind land-use development and facilitating land management policies.

The aim of current study is to map and explain the land-use changes in Lithuanian municipalities in the period since 1971. We map land use types that are considered the most significant in terms of carbon storage using land-use data originating from the Lithuanian NFI. Then, we evaluate and explain the land-use changes during different periods using factors that are extracted from freely available GIS databases. Finally, we discuss the spatial patterns observed in both land use and land-use change geography, associating them with land-use policy implications.

## 2. Materials and Methods

### 2.1. Study Area

The study focuses on land use and land-use changes in Lithuania. Geographically, even though Lithuania is situated in central Europe with central coordinates of 55°10′ N, 23°39′ E (Figure 1), it has strong historical links with Eastern Europe. The total land area of Lithuania is 65,200 km². Lithuania lies on the Eastern European Plain, with characteristic lowlands and hills (the highest point in the country is only 293 m above sea level). The terrain features numerous lakes and wetlands, and a mixed forest zone covers over 33% of the country. Lithuanian climate conditions and natural soil productivity are generally favourable for crop production. Consequently, more than 50% of its land area is used for agricultural purposes. Currently, Lithuania is dominated by rural landscapes, covering approximately 75% of its territory. The proportion of natural landscapes does not exceed

15% of the country's area and they are concentrated in the eastern and southeastern regions, the hilly western parts of the country, and the ancient delta on the shoreline [44]. The rest of the country is covered by rapidly expanding urban or urbanized landscapes. The administrative units of the Republic of Lithuania are 10 counties and 60 municipalities.

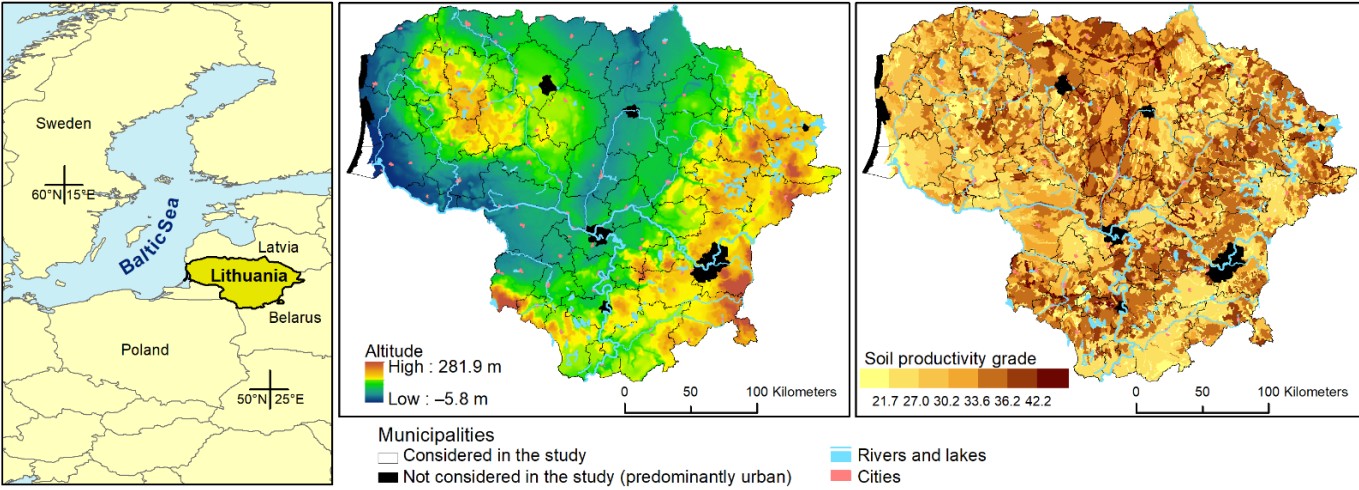

**Figure 1.** Specification of the study area: **left**—location of the study area in Europe, **centre**—elevation in Lithuania, **right**—average soil productivity grade. Sources of the data used: **left**—thematicmapping.org/downloads/world_borders.php (accessed on 13 May 2021), **centre**—GDB200 database from www.gis-centras.lt/ (accessed on 13 May 2021), **right**—derived using Dirv_DR10LT from www.geoportal.lt (accessed on 13 May 2021).

### 2.2. Input Data

Two types of input data were used in the study—(i) data describing the land uses in Lithuania and (ii) data describing the factors influencing the land-use changes. Land-use information was available from the Lithuanian National Forest Inventory, which involves permanent observation of land-use types on a network of 16,349 systematically distributed sampling plots [36,45]. NFI sampling plots are distributed in all land-use types across the country in clusters of four sampling plots on a 4 × 4 km grid. One-fifth of the sampling plots are visited each year by field measurement specialists; therefore, the whole country is covered in a five-year inventory cycle. Land-use types and subtypes are identified annually at the centre of each plot from 1971 using the nomenclature of GHG inventories [46], and land-use changes, if occurring, are detected and reported according to the measurement year. The land cover is further grouped according to the GHGC Level 1 coding of land cover: forest, producing land, grassland/pasture, wetlands, built-up areas, and other land. It should be noted that the identification and monitoring of land-use types became the responsibility of the Lithuanian NFI in 2011. To reconstruct the land-use types for each of the nonforest sampling plots for the period 1990–2011, a special study was conducted based on the use of all available historical materials, e.g., remotely sensed data, including orthophotos and satellite image archives, and land management and real estate maps [37].

Land-use statistics were aggregated to the level of Lithuanian municipalities. The borders of municipalities (USE_3 level) were acquired from EuroBoundaryMap (v3.0), which is a European reference database of administrative units and boundaries established within the framework of EuroGeographics (Available online: eurogeographics.org/maps-for-europe/ebm/, accessed on 13 May 2021). We excluded from the study nine predominantly urban municipalities (Figure 1); thus, the study was done on 51 municipalities with a mean area of 1260 km$^2$ (standard deviation = 452). The municipality for each observation point was identified using the Spatial Join tool of ArcGIS (v10.7) by specialists of the State Forest Service responsible for GHG inventories in the LULUCF sector. Summarized data on all the land-use types and subtypes from 1971 to 2015 were joined to the borders of

each municipality. Usually, the proportions of observation points belonging to particular land-use types were calculated for each municipality and used in further analyses.

Free data available from the Spatial information portal of Lithuania (Available online: geoportal.lt, accessed on 13 May 2021) were used to describe the factors influencing the land-use changes. The datasets used to get the explanatory variables were the Georeference spatial dataset (GDR10LT), a soil spatial dataset at a scale of 1:10,000 (Dirv_DR10LT), a land reclamation and wetness dataset at a scale of 1:10,000 (Mel_DR10LT), a dataset of special land-use conditions at a scale of 1:10,000 (SŽNS_DR10LT), a dataset of abandoned agricultural land (AŽ_DRLT), CORINE land covers for 1995, 2000, 2006 and 2014, a land parcel block database referring to 2004, 2008 and 2014 (KŽS), population census data for 1970, 1989, and 2011, including geospatial data for 2011, data on agricultural crops declared to the National Paying Agency for 2010–2015, and a digital raster elevation model (cell size: 100 m) built based on information available in the GDB200 GIS database. Each vector dataset was overlain with the municipality polygons and summary statistics, such as total area or length, and the area/count proportion was extracted for a specific geographic object or phenomenon. If the explanatory variable was available in the raster, we used ArcGIS function Zonal Statistics to estimate the statistics of a certain attribute within each municipality. In the case, the geographical data required additional processing, so the standard functionality of ArcGIS Desktop was used. In such a way, e.g., the slope was estimated using the digital elevation model as the input. To estimate the population within a 15-min driving distance of the centre of each municipality, we used a road database, referring to the year 2007. The road network was constructed using input vector data corresponding to current data of the Georeference background cadastre (GRPK), with all field and forest roads included. Accessibility was calculated using standard ArcGIS Network Analyst New Service Area functionality within the framework of the FP7 RURALJOBS project [47]. Additionally, we used agricultural census data, available from the Official Statistics Portal of Lithuania [48]. All the attributes characterising the municipalities are summarised in Table A1.

### 2.3. Mapping and Evaluating the Land-Use Spatial Pattern

The proportions of forest, producing land, meadow/pasture, wetlands, built-up land, and other land in municipalities were plotted on the map. The Global Moran's *I* statistic and Anselin Local Moran's *I* were used to identify global and local patterns in the distribution of land-use characteristics in Lithuanian municipalities, respectively. To estimate the spatial distribution patterns, we used the spatial statistics tools available in ArcGIS Desktop. The land uses in municipalities were visualized and analysed at the following points: 1971, 1990, 2005, and 2015. The first and last years refer to the starting and ending points of land-use data available for the study, and the years 1990 and 2005 were chosen to correspond to the restoration of Lithuanian independence and joining the European Union, respectively. These dates also fit the overall development trajectories of producing land and meadow/pasture for the whole country [22]. To quantify the presence of a monotonic increasing or decreasing trend in the changes of land-use proportions during a specific period, we performed a nonparametric Mann–Kendall test and then estimated the slope of the linear trend with the nonparametric Sen's method using MAKESENS tools [49]. The spatial distribution of the slope was visualized and analysed using the same approaches as used with the land-use proportions and described above. The trends were analysed for the following periods: 1971–2015, 1971–1990, 1990–2005, and 2005–2015.

To understand the factors behind the land-use changes in Lithuanian municipalities, we applied an ordinary least squares (OLS) regression. The focus was on the changes in proportions of forest, producing land, and meadow/pasture during all the periods mentioned above. As the dependent variable, the slope of the linear trend in land-use proportion changes was used. All the variables extracted from the freely available GIS databases were considered as candidates for explanatory or independent variables. We checked all possible combinations of input candidate explanatory variables using the Exploratory Regression

tool of ArcGIS Desktop. The number of independent variables ranged from two to five. The following conditions for the fit of the regression models were set: only explanatory variables with statistically significant coefficients (95% confidence level) and with a variance inflation factor under 7.5 were exploited to avoid multicollinearity; the minimum Jarque–Bera *p*-value was 0.1 to consider the model residuals to be normally distributed; and model residuals were tested for spatial clustering using Global Moran's *I* (maximum value allowed: 0.1) for the cases that met all the above search criteria. We evaluated the extent to which each candidate independent variable met the above conditions. Only the best regression models (in terms of adjusted $R^2$ and corrected Akaike information criterion, under the condition that all other statistical tests—Jarque–Bera statistic, Koenker (BP) statistic, variance inflation factor, and spatial autocorrelation of the regression residuals—were passed) are presented in the current paper.

The methodological framework of our study is summarized in Figure 2.

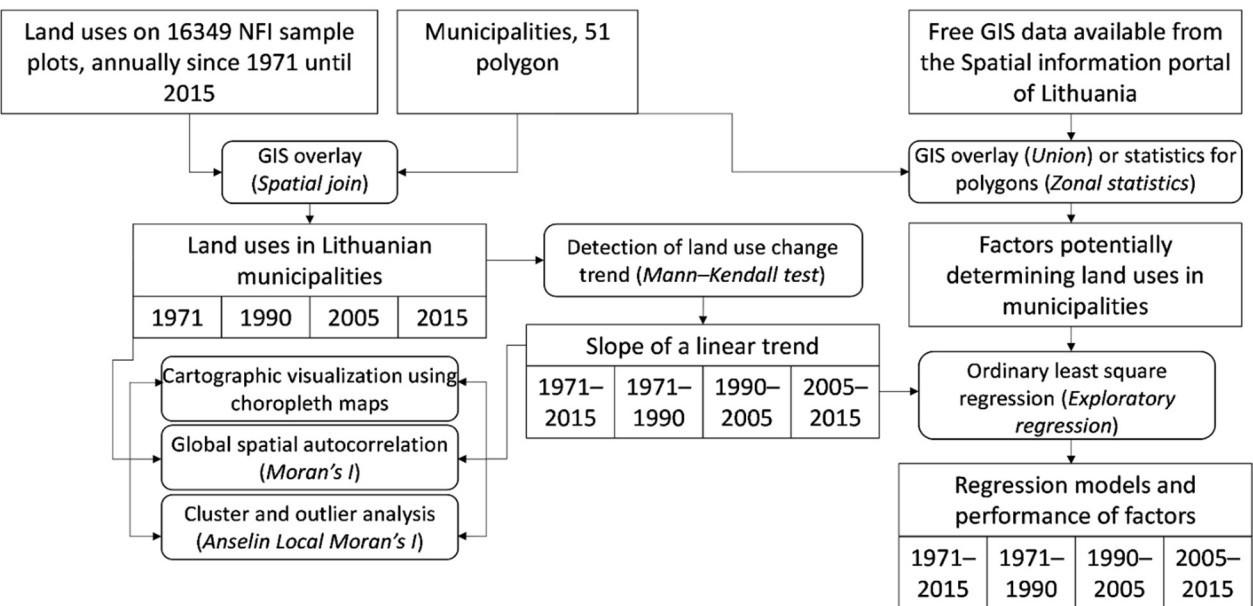

**Figure 2.** Flowchart summarizing the overall structure of the study.

## 3. Results

Agricultural landscapes dominate in Lithuania. Land-use types contributing the most to the carbon accumulation in the LULUCF sector (forest, producing land, and meadow/pasture) covered, in 1971, rather similar proportions of the country's area—each around 28–30%. The areas of other land uses accounted for less than 12%. Even though Lithuania is a small country, the land-use proportions in different parts of the country differed. Additionally, if taking into consideration only two years, i.e., 1971 and 2015, one could state that the areas of forest, producing land, and built-up land did increase, while the proportion of meadow/pasture, wetlands, and other land decreased. However, the trajectories of specific land-use development during shorter periods experienced notable changes.

Even though there is no statistically significant global autocorrelation in values of forest proportion in Lithuanian municipalities, the southeastern and western parts of the country are more forested (Figure 3). Lower forest proportions are found in northern and central municipalities, where producing land dominates. The global spatial autocorrelation of agricultural land proportions in Lithuanian municipalities—both producing land and pasture/meadow—was statistically significant at practically all the time points used for the analysis. Producing land dominated in the northern and central municipalities, with lower forest proportions. Larger proportions of pasture/meadow were reported in municipalities

with higher forest proportions, but not along the southeastern border of the country with overall forest dominance. The proportions of other land uses in Lithuanian municipalities are notably lower and usually do not exhibit global spatial autocorrelation. The Anselin Local Moran's *I* statistic was used to explore the spatial clusters of features with high or low values, as well as the spatial outliers. Two clusters of municipalities with low proportions of forest area that were stable over time and neighboured by municipalities with low values were identified. They practically overlapped with the high–high clusters of producing land abundance. It should be noted that the high–high cluster of producing land proportion in the northern part of Lithuania was the highest one among all clusters identified in this study, made up of 4–7 municipalities. This cluster also overlapped with the low–low cluster of pasture/meadow. Municipalities in the eastern part of Lithuania made up the low–low cluster of producing land proportions, which partly overlapped with a high–high cluster of wetlands that was stable over time. A high–high cluster of meadow/pasture was identified in the western part of the country, in the lowland associated with the Nemunas Delta area. Spatial outliers were usually small, i.e., including just one municipality and associated with municipalities with forest proportions that were different from their neighbourhoods. Local clusters of proportions of built-up areas were also small and dispersed throughout the whole country. Local spatial clusters and outliers of other land exhibited rather random occurrence patterns over time; however, the low proportions of that land-use type in the municipalities should be kept in mind.

The areas of forest and built-up land increased in Lithuania since 1971, while the areas of producing land, pasture/meadow, wetlands, and other land went down—this is suggested by, respectively, the positive and negative values of the slope of the linear trend (Table 1). Stable development trajectories were followed by the proportions of forest, wetland, built-up land, and other land during the whole period under assessment; however, the areas of producing land and pasture/meadow did both increase or decrease during specific periods. Thus, the areas of producing land were increasing at the cost of a decrease in pasture/meadow from 1971 to 1990. By the end of this period, the area of producing land was at its highest level—36%. The area of producing land decreased since 1990, with the proportion of pasture/meadow increasing to be level with the areas of key agricultural land-use types in 2005, at a level of 28%. Finally, the trajectories as they were since 1971 were repeated after 2005.

**Table 1.** Trends of change in proportion of land-use types across the whole of Lithuania (significance level of slope: ***, 0.001; **, 0.01; and *, 0.05).

| Land-Use Type | Trend Statistics for the Period under Review | | | | | | | |
|---|---|---|---|---|---|---|---|---|
| | 1971–2015 | | 1971–1990 | | 1990–2005 | | 2005–2015 | |
| | Slope | Z Statistic | Slope | Z Statistic | Slope | Z Statistic | Slope | Z Statistic |
| Forest | 0.085 | 9.67 *** | 0.076 | 6.13 *** | 0.064 | 5.36 *** | 0.106 | 4.20 *** |
| Producing land | −0.027 | −0.69 | 0.539 | 5.09 *** | −0.624 | −5.36 *** | 0.418 | 4.05 *** |
| Grassland/pasture | −0.031 | −0.78 | −0.579 | −5.16 *** | 0.612 | 5.36 *** | −0.542 | −4.05 *** |
| Wetlands | −0.020 | −9.52 *** | −0.023 | −6.10 *** | −0.012 | −4.95 *** | −0.007 | −3.74 *** |
| Built-up land | 0.009 | 6.79 *** | −0.001 | −2.98 ** | 0.014 | 4.95 *** | 0.015 | 3.97 *** |
| Other land | −0.015 | −7.47 *** | −0.002 | −2.37 * | −0.016 | −4.86 *** | 0.001 | 0.93 |

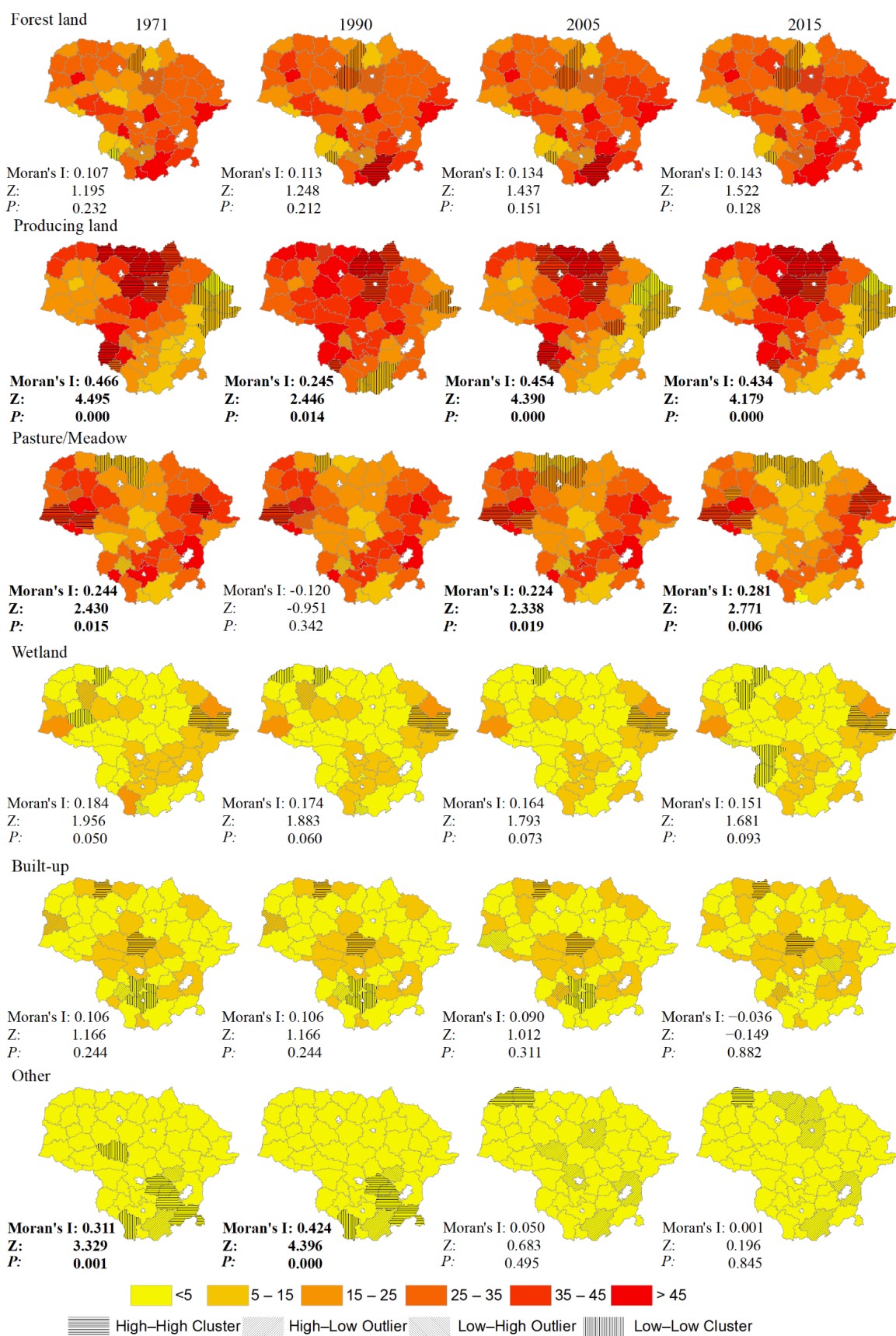

**Figure 3.** Area proportions of land-use types in Lithuanian municipalities during different periods since 1971. Statistically significant values of Global Moran's *I* statistic are in bold. Linear shades identify statistically significant hotspots, cold spots, and spatial outliers based on the Anselin Local Moran's *I* statistic.

Furthermore, the spatial patterns of changes in three land-use types in Lithuanian municipalities were analysed, i.e., forest, producing land, and pasture/meadow, and are presented in Figure 4. The slope of the linear trend of forest proportion changed both during the whole period (1971–2015) and in all three shorter spans in an interval between –0.5 and 0.5, suggesting rather slow development. Statistically significant global spatial autocorrelation in the slope values was observed only for 1971–1990. Even though the slope values were low, there were some spatial clusters and outliers identified, such as the low–low cluster suggesting aggregation of municipalities with decreasing forest proportion during 1971–2015 in the central part of the country and some southwestern municipalities since 1990 or the high–high cluster in 1971–1990 in municipalities along the border of the former Soviet Union and Poland. The slope of a linear trend for the development of forest proportion in 1971–2015 was statistically significant in practically all the municipalities. However, if shorter periods were taken into consideration, usually only positive slope values were statistically significant at the level of the municipality. The trends of producing land changes in the municipalities were inverse to the ones of pasture/meadow. This refers both to the value of the slope of linear trend and the types and the location of spatial clusters. The areas of producing land increased most intensively in 1971–1990 in the eastern and western parts of the country, resulting in statistically significant global spatial autocorrelation and local spatial clusters. However, since the restoration of independence in Lithuania in 1990, the proportion of producing land started to decrease, with the most intensive drop in the municipalities, where the increase was faster before 1990. Opposite trends could be reported for the development of pasture/meadow. Finally, since 2005, agricultural land uses changed their trajectories once again. Even though there is no statistically significant global autocorrelation in the value of the slope for the proportion of producing land—the area of this land-use type was increasing practically all municipalities, with some small spatial clustering effects—the decrease in pasture/meadow was faster in the central part of Lithuania (with the highest global Moran's *I* statistic among all the cases estimated). If the whole period of 1971 to 2015 is taken into consideration, the value of the linear slope for producing land and pasture/meadow was usually statistically nonsignificant for most of the municipalities, suggesting large fluctuations in land-use type proportions over the time. However, if taking into consideration shorter periods, the slope of linear trend was statistically significant in the majority of municipalities—e.g., for 1971–1990, there were just six municipalities with nonsignificant slope values for both producing land and pasture/meadow, or 10 and eight municipalities, respectively, for the period 1990–2005.

To explain the land-use change trends in Lithuanian municipalities during different periods of the last half-century, we used information available from different GIS databases and multiple linear regression. If taking into consideration the whole period (1971–2015), the best explained variable was the slope of steadily increasing forest proportion (Table 2). The best regression models explained 65% of the variance of the slope of forest proportion changes. The figures for producing land and pasture/meadow were, respectively, 40% and 37%. When considering a shorter period, the percentage of variance explained by forest change models decreased but increased in models for meadow and pasture. In the case of producing land, the coefficient of determination only increased in 1971–1990.

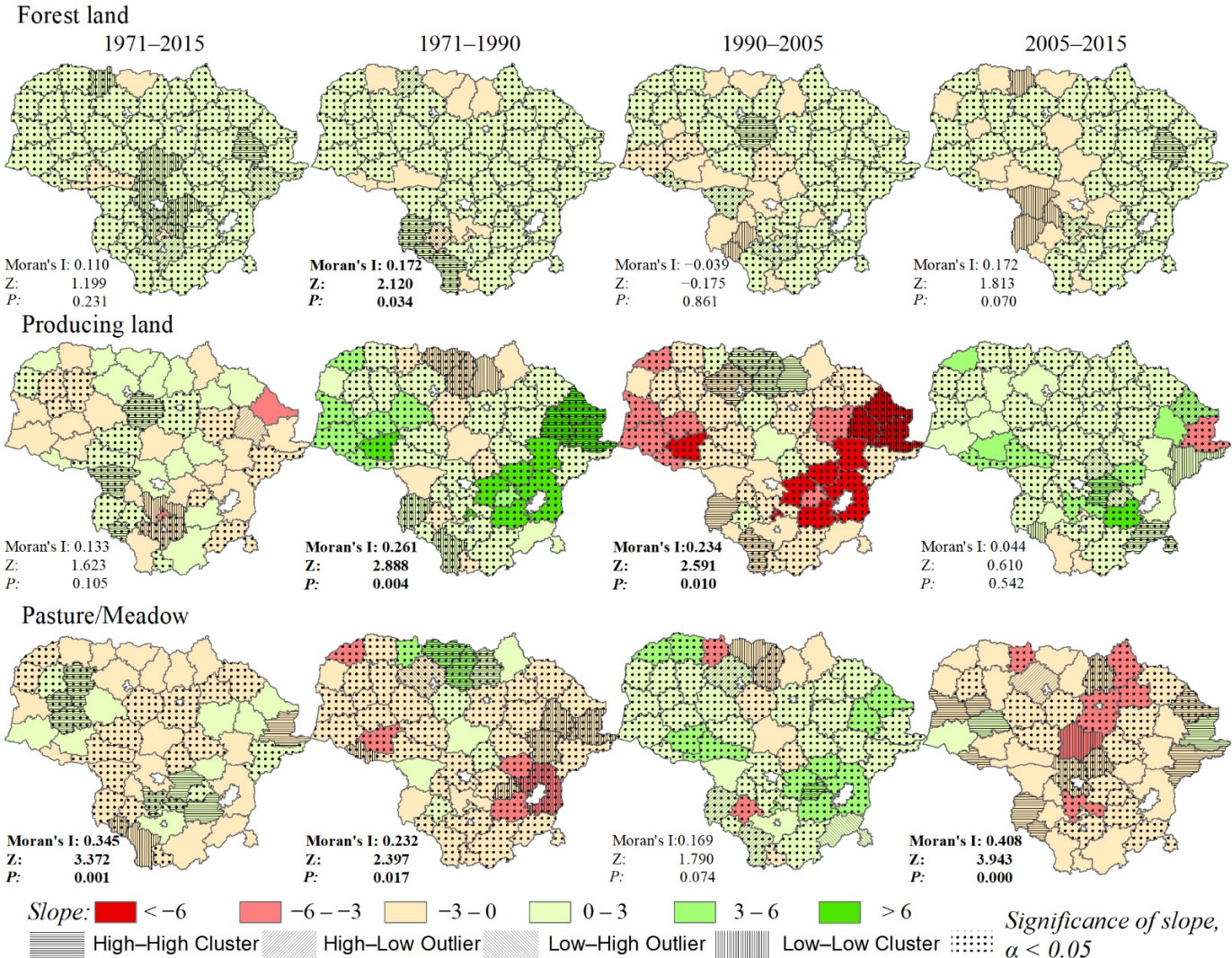

**Figure 4.** Slope of linear trend in changes of area proportions of main land-use types in Lithuanian municipalities during different periods since 1971. Statistically significant values of Global Moran's *I* statistic are in bold. Linear shades identify statistically significant hotspots, cold spots, and spatial outliers based on the Anselin Local Moran's *I* statistic. Dotted areas identify the statistical significance of the slope in a certain municipality.

The proportion of the time that each candidate explanatory variable was detected to be statistically significant, testing all potential combinations of variables, is illustrated in Figure 5. Usually, there were more variables with larger significance when modelling the change trends of forest. The abundance of land-use types in the beginning of each analysed period was among the most significant factors in most of the tested cases. Forest proportion in the municipality also had an impact on the development trends of other land uses. Soil productivity was another factor often present in the models. Terrain-related attributes played a more important role in forest and grassland change models. Topographic details participated in forest change models. It should be noted that the impacts of explanatory variables were similar in the forest and grassland change models but opposite when modelling the producing land development.

**Table 2.** Characteristics of best multiple linear regression models for each analysed dependent variable over different time periods.

| Adjusted $R^2$ | Corrected Akaike Information Criterion | Jarque–Bera Statistic | Koenker (BP) Statistic | Variance Inflation Factor | Moran's $I$ of the Regression Residuals | Model |
|---|---|---|---|---|---|---|
| | | | | Period: 1971–2015 | | |
| | | | Dependent Variable: Slope of Linear Trend of Forest Proportion Changes in Lithuanian Municipalities | | | |
| 0.65 | −64.94 | 0.68 | 0.21 | 1.60 | 0.39 | 1.474085 − 0.004053 × [Population density, 2011] ** − 0.02052 × [Soil productivity grade] *** + 0.003441 × [Standard deviation of altitude] − 0.010011 × [Forest, 1971] *** |
| | | | Dependent Variable: Slope of Linear Trend of Producing Land Proportion Changes in Lithuanian Municipalities | | | |
| 0.40 | 111.9 | 0.00 | 0.00 | 5.02 | 0.20− | 0.39185 + 0.032059 × [Standard deviation of altitude] * − 5.077213 × [Mean slope] ** + 3.110821 × [Standard deviation of slope] ** − 0.395311 × [Grassland area per cattle-unit, 2014] * |
| | | | Dependent variable: slope of linear trend of grassland proportion changes in Lithuanian municipalities | | | |
| 0.37 | 43.33 | 0.12 | 0.55 | 4.89 | 0.40 | −0.722898 + 1.852514 × [Mean slope] *** − 1.465708 × [Standard deviation of slope] ** + 0.012207 × [Grassland, 1971] *** + 0.000001 × [Protected areas] ** |
| | | | | Period: 1971–1990 | | |
| | | | Dependent Variable: Slope of Linear Trend of Forest Proportion Changes in Lithuanian Municipalities | | | |
| 0.40 | 24.99 | 0.00 | 0.11 | 3.25 | 0.22 | 0.484 − 0.0049 × [Land reclamation intensity] + 0.003281 × [Minimum altitude] + 0.012543 × [Standard deviation of altitude] ** − 0.013414 × [Forest, 1971] *** |
| | | | Dependent Variable: Slope of Linear Trend of Producing Land Proportion Changes in Lithuanian Municipalities | | | |
| 0.45 | 322.44 | 0.00 | 0.04 | 5.83 | 0.82 | −3.213562 + 0.826977 × [Soil productivity grade] ** − 0.1465 × [Land reclamation intensity] ** − 0.232591 × [Forest, 1971] *** − 0.433926 × [Producing land, 1971] *** |
| | | | Dependent Variable: Slope of Linear Trend of Grassland Proportion Changes in Lithuanian Municipalities | | | |
| 0.58 | 175.65 | 0.00 | 0.43 | 2.24 | 0.68 | 4.890114 − 0.009301 × [Range of altitude] * + 2.125683 × [Mean slope] − 0.070913 × [Forest, 1971] *** − 0.113908 × [Grassland, 1971] *** |
| | | | | Period: 1990–2005 | | |

**Table 2.** *Cont.*

| Adjusted $R^2$ | Corrected Akaike Information Criterion | Jarque–Bera Statistic | Koenker (BP) Statistic | Variance Inflation Factor | Moran's $I$ of the Regression Residuals | Model |
|---|---|---|---|---|---|---|
| \multicolumn{7}{c}{Dependent Variable: Slope of Linear Trend of Forest Proportion Changes in Lithuanian Municipalities} | | | | | | |
| 0.42 | −41.24 | 0.36 | 0.08 | 2.34 | 0.45 | 1.686125 − 0.032764 × [Soil productivity grade] *** − 0.007497 × [Standard deviation of altitude] *** − 0.006543 × [Forest, 1990] *** + 0.000001 × [Area of agricultural blocks, 2004] *** − 0.000001 × [Area of water bodies, 2004] ** |
| \multicolumn{7}{c}{Dependent Variable: Slope of Linear Trend of Producing Land Proportion Changes in Lithuanian Municipalities} | | | | | | |
| 0.34 | 349.86 | 0.00 | 0.01 | 2.62 | 0.80 | −8.849178 + 0.277094 × [Land reclamation intensity] *** − 0.190026 × [Producing land, 1990] * − 0.000001 × [Protection zones of electricity lines] + 0.000001 × [Protected areas] ** − 0.000001 × [Area of water bodies, 2004] * |
| \multicolumn{7}{c}{Dependent Variable: Slope of Linear Trend of Grassland Proportion Changes in Lithuanian Municipalities} | | | | | | |
| 0.45 | 198.44 | 0.44 | 0.55 | 2.34 | 0.68 | 3.864368 − 0.072516 × [Land reclamation intensity] *** − 0.047102 × [Minimum altitude] *** + 0.018176 × [Range of altitude] *** − 0.000034 × [Population < 15-min drive to cities] ** + 0.000001 × [Area of water bodies, 2004] *** |
| \multicolumn{7}{c}{Period: 2005–2015} | | | | | | |
| \multicolumn{7}{c}{Dependent Variable: Slope of Linear Trend of Forest Proportion Changes in Lithuanian Municipalities} | | | | | | |
| 0.47 | 12.58 | 0.33 | 0.08 | 2.04 | 0.88 | −0.427383 + 0.003047 × [Mean slope] *** − 0.014434 × [Standard deviation of altitude] *** + 0.017336 × [Grassland, 2005] *** + 0.000001 × [Area of agricultural blocks, 2008] *** − 0.000001 × [Area of built-up blocks, 2008] ** |
| \multicolumn{7}{c}{Dependent Variable: Slope of Linear Trend of Producing Land Proportion Changes in Lithuanian Municipalities} | | | | | | |
| 0.29 | 194.65 | 0.12 | 0.01 | 4.86 | 0.75 | 1.326413 − 0.025171 × [Minimum altitude] ** − 0.000049 × [Area of agricultural blocks, 2008] ** + 0.000042 × [Private land area, 2008] ** + 0.058605 × [Grassland, 2005] *** − 0.000001 × [Area of water bodies, 2008] ** |
| \multicolumn{7}{c}{Dependent Variable: Slope of Linear Trend of Grassland Proportion Changes in Lithuanian Municipalities} | | | | | | |
| 0.50 | 140.2 | 0.56 | 0.37 | 4.66 | 0.13 | 4.202848 − 0.198318 × [Soil productivity grade] *** + 0.637596 × [Grassland area per cattle-unit, 2008] ** + 0.063401 × [Forest, 2005] *** − 0.000001 × [Length of streams, 2014] ** + 0.000001 × [Area of water bodies, 2014] ** |

Note: The statistical significance of each coefficient in the model is noted as follows: *, $p = 0.10$; **, $p = 0.05$; ***, $p = 0.01$.

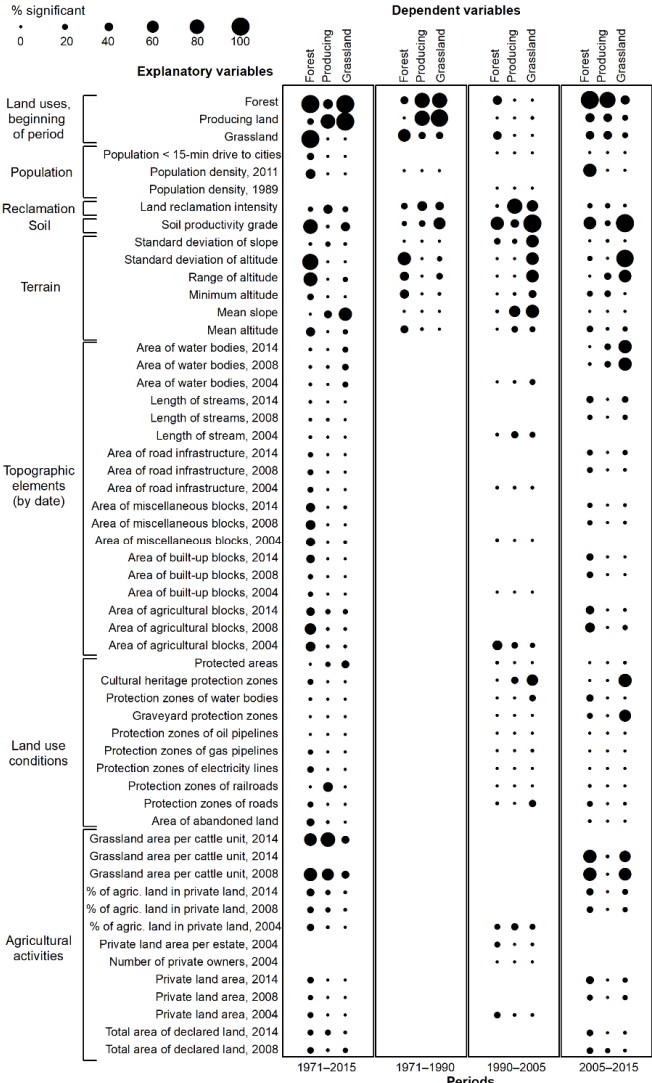

**Figure 5.** The proportions of times that each candidate explanatory variable was statistically significant when testing all potential regression models. Blank means that the variable was not included in the exploratory regression runs.

## 4. Discussion

The overall trend in Lithuania during the last five decades has been increases in the areas of forest and built-up lands and decreasing areas of producing land, meadow/pasture, wetlands, and other land uses. Nevertheless, the development trends for the proportions of producing land and meadow/pasture changed their trajectories several times. The breakpoints in the development of key agricultural land uses were linked with important dates in Lithuanian history. This suggests that the land-use development trends could be impacted by political processes in and around the land management and use relationships. Three periods were singled out with potentially differing land-use conditions. The first period (1971–1990) we associate with the development of large agricultural enterprises under the condition of a planned economy, as Lithuania was one of the former Soviet Union republics. Restructuring of agriculture started in 1991–1992. The reform of the national agrarian sector took place since the restoration of independence, which resulted in introducing private land ownership, together with changed overall principles of agriculture and land management. This was followed by a period of European Union and state budget support allocated to agriculture and rural development, since Lithuania joined the EU in 2004 [50–52]. The stable overall increase in forest could be explained by command-and-

control forest governance restricting radical changes, strict deforestation control, and the aspiration to preserve domestic forest resources [22,53,54]. Thus, the trends observed in our study are associated, first of all, with political and social factors rather than natural conditions. This is supported by Ribokas and Milius [55], who argued that nearly all legal, economic, and social land management reforms in Lithuania were neither consistent nor unambiguous.

Even though Lithuania is a relatively small country with rather smoothly changing geographic conditions, we could still observe statistically significant patterns in the land-use distribution and changes. The increase in forest was largest in southwest Lithuania, potentially due to the fast increase during 1971–1990. Since 2005, however, forest increased the most in northeastern Lithuania and the hilly municipalities in the western part of the country. We explain this by the intensive afforestation of abandoned land or land not used for agriculture. The trajectories of producing land development were different during the periods analysed. If taking into consideration the last five decades, the overall decrease in producing land in the hilly areas of western and eastern Lithuania could be explained by the fast decrease in producing land in 1990–2005. These areas are less favourable for agriculture, and the presence of abandoned agricultural land is more common here. However, the development of producing land proportion was radically different in these areas during other periods, i.e., 1971–1990 and 2005–2015. Development trajectories of meadow/pasture were, at least in principle, the opposite to those of producing land. The most rapid reduction in meadow/pasture during the whole period analysed was in the flat central and northern municipalities with the most fertile soil for agriculture. The fastest decrease in meadow/pasture was seen here since 2005. Usually, producing land is converted into meadow/pasture, and vice versa. Similar changes were also noted by Aleknavičius [56], who reported that the area of producing land in Lithuania decreased by, on average, 18,900 ha annually in 1948–1989 and by even more—51,800 ha—during 1990–2005, with large areas of producing land converted into meadow/pasture. The total area of agricultural land was reported to have shrunk by 2.35% during 2007–2017 [57]. The decreasing area of agricultural land was explained by increasing forest and new housing areas, especially in hilly western regions [58,59]. The forest area of Lithuania is reported to have increased during the period since 1950 [41,60]. Usually, the largest increase in forest proportion is found in regions least favourable for agriculture. The largest areas of new forests emerged in southeastern Lithuania, while the slowest increase om forest was in the least forested municipalities. Some forest loss was also reported [41] since the 1950s, associated with forest transformation into agricultural land, or less frequently into scrubland or water bodies. The latter transformation was related with the construction of large artificial reservoirs. It should be noted that all of the national studies mentioned above, except for Juknelienė and Mozgeris [41], did not use spatial statistics to support their findings on land-use distribution patterns. Similar forest and agricultural land changes were reported in neighbouring countries, e.g., in Poland [61].

The available land use and land-use change patterns are usually associated with interactions between socioeconomic and cultural land management conditions, biophysical constraints, and land-use history [62]. To specify the interactions, we have chosen the multiple regression. Our focus in the current study was on the characterization—or, at least, identification—of the most important biophysical and socioeconomic drivers of land use in Lithuania. Usually, the candidate drivers are suggested based on a literature review and expert knowledge. We introduced one extra criterion: the driver needs to be described using easily available data. In addition to census data, we gathered study information available from the Spatial Information Portal of Lithuania. The majority of such spatial information was captured during the last few decades; thus, this could have impacts on the performance of the regression models developed for the earlier periods covered in our study. The best regression model, in terms of $R^2$, was developed to explain the changes in forest proportion during the whole period (i.e., 1971–2015). However, the development of forest was very smooth during the whole period. Shorter periods resulted

in better performance of the regression models if modelling the proportion changes of meadow/pasture and, partly, the proportion of producing land. In all cases, the Akaike Information Criterion values for models with a shorter time period were higher than those for the land-use change from 1971 to 2015. In addition to the availability and quality of historical explanatory driver variables, multiple regression in land-use change analyses can be used for relatively short time periods, i.e., one or two decades [63]. We should also emphasise that we did not aim to elaborate the overall best regression models, i.e., the focus was on testing all potential driver variables in all potential combinations, taking into consideration, of course, the statistical significance and multicollinearity of factors and properties of model residuals.

If taking a closer look at the performance of each tested candidate driver variable, the importance of the forest proportion at the beginning of each period stands out. We could consider the abundance of forest in the municipality as a key indicator of landscape stability [64]. In 2019 forest covered 33.7% of Lithuania [60], and a political objective was set to increase this figure to at least 35% by the year 2030 [65]. Assuming that the annual forest area increase rate during the period from 1971 until 2015 was 0.085% (0.108% during the last decade), this objective could be achieved by increasing the country's forest area by at least 0.118% per year. This challenging task would impact the development of other land uses, both considering the models suggested in the current study and the practice of afforesting abandoned or unsuitable agricultural land [65]. We identified the soil productivity grade as an important factor shaping land-use changes, even though there was some scepticism regarding using the crop production potential of the land for exploring land-use change patterns [66,67]. Soil productivity grade was most strongly correlated with the change trends of producing land and meadow/pasture proportion (Table 3). It was a statistically significant contributor in models explaining, e.g., forest changes (the factor was significant in 69% and 61% of all cases tested for the periods 1990–2005 and 2005–2015, respectively) and grassland changes (98% and 97%). Population is usually reported as an important factor influencing land-use distribution [68–74]. We did not directly use the statistics on, e.g., the ratio between the urban and rural population; however, we integrated the factors that were used to specify the rural population in the recent FP7 RURALJOBS project [47]. However, neither population density nor the share of population within a specified driving distance of cities was found to be among the most important factors. The reason could also be the reference date of the population data—e.g., the population density in 2011 was a significant factor in nearly 70% of cases tested to describe forest area changes after 2005. Land reclamation is considered an important factor that has been shaping Lithuanian landscapes in the second half of the 20th century [75–77]. It should be emphasized that the facilities available for land reclamation in Lithuania influence the land use—e.g., afforestation of agricultural lands, is dependent on the presence or absence of land with a functioning land reclamation system [78]. In our study, the intensity of land reclamation in the municipality is an important factor for explaining changes in producing land and meadow/pasture. The topography of the landscape is usually closely related to the land use and land-use change patterns [62,79,80]. However, this attribute is scale-dependent; thus, relatively coarse-scale elevation data sources were used to reveal the general trends. Even though Lithuania can be characterised as a lowland country (cf. Figure 1), there are differences in the land use and land-use change patterns observed between the hilly and relatively flat municipalities. Topography-related factors are, therefore, more effective at explaining changes in agricultural land. In Lithuanian municipalities, the soil productivity is inversely correlated with the average altitude ((Pearson's correlation coefficient $-0.579$ ($N = 51$))), slope steepness (Pearson's correlation coefficient $-0.552$ ($N = 51$)), and diversity of elevation conditions, expressed as a standard deviation of altitude (Pearson's correlation coefficient $-0.333$ ($N = 51$)) or slope steepness (Pearson's correlation coefficient $-0.510$ ($N = 51$)). The land-use change transitions usually involve conversion from producing land into meadow and pasture or vice versa, usually on land less suitable for growing crops.

**Table 3.** Pearson's correlation coefficient between a selected explanatory variable and the slope of the linear trend in the development of a specific land-use proportion over a certain period (*N* = 51).

| Selected Explanatory Variable | Forest | | | | Producing Land | | | | Meadow/Pasture | | | |
|---|---|---|---|---|---|---|---|---|---|---|---|---|
| | 1971–2015 | 1971–1990 | 1990–2005 | 2005–2015 | 1971–2015 | 1971–1990 | 1990–2005 | 2005–2015 | 1971–2015 | 1971–1990 | 1990–2005 | 2005–2015 |
| Soil productivity grade | −0.347 | −0.247 | −0.202 | −0.337 | 0.314 | −0.427 | 0.408 | −0.175 | −0.407 | 0.609 | −0.520 | −0.580 |
| Population density in 2011 | −0.295 | −0.117 | −0.284 | −0.384 | 0.064 | −0.047 | 0.014 | −0.014 | −0.135 | 0.067 | −0.094 | −0.061 |
| Land reclamation intensity | −0.091 | −0.216 | 0.032 | −0.178 | 0.426 | −0.511 | 0.502 | 0.006 | −0.364 | 0.506 | −0.398 | −0.387 |
| Standard deviation of altitude | 0.421 | 0.474 | 0.009 | −0.043 | −0.033 | 0.132 | −0.134 | 0.117 | 0.121 | −0.332 | 0.365 | 0.441 |
| Mean slope | 0.123 | 0.298 | −0.015 | 0.025 | −0.458 | 0.393 | −0.410 | 0.096 | 0.409 | −0.450 | 0.420 | 0.306 |
| Forest | −0.499 | −0.259 | −0.106 | −0.121 | −0.231 | 0.136 | 0.567 | −0.037 | 0.127 | −0.270 | −0.590 | 0.049 |
| Producing land | −0.130 | −0.094 | −0.118 | −0.257 | 0.468 | −0.290 | 0.572 | −0.079 | −0.512 | 0.348 | −0.628 | −0.628 |
| Grassland | 0.577 | 0.551 | 0.397 | 0.424 | −0.385 | 0.042 | −0.541 | 0.185 | 0.515 | 0.016 | 0.597 | 0.623 |

## 5. Conclusions

The annual land-use changes in Lithuanian municipalities were identified for the period 1971–2015 using sampling-based information from the Lithuanian National Forest Inventory. Originally developed to support strategic forest planning with data, the Lithuanian NFI was recently adopted to monitor land-use changes. We demonstrate its usability to explore land use and land-use change properties. Lithuania, being a relatively small lowland country, exhibits statistically significant spatial patterns in land use and land-use change distribution. Since 1971, the area of land uses important for carbon storage (forest, producing land, and meadow/pasture) was similar—20–37% each. Since then, the proportion of producing land, forest, and built-up areas did increase, while the proportions of meadows and pastures, wetlands, and other lands went down. The area of forest, wetlands, built-up areas, and other land changed relatively steadily over the last five decades. However, the trends of changes in producing land and meadow/pasture depended on the historical period, being associated with historical periods impacted by political processes in and around land management and use relationships. The proportions of producing land and pasture/meadow remained spatially autocorrelated during the whole period analysed. Local spatial clusters and outliers were identified for all land-use types at each time point analysed, suggesting the need for spatially explicit land-use management policies.

Exploiting the information from publicly available GIS and agricultural census databases, we managed to explain, using multiple linear regression, up to 65% of the variance in forest, 40% in producing land, and 37% in meadow/pasture proportion changes over the entire period of 1971–2015. The regression models usually improved with shorter time periods for producing land and meadow/pasture proportion changes. Usually, the factors shaping the changes in the proportions of forest and meadow/pasture were similar, but different from those affecting producing land changes. We associated the trends in land-use changes and the models explaining them with the interactions of political, natural, and social systems.

We also conclude that a spatially explicit assessment of the land-use pattern can identify critical areas of land-use change and give insight to improve land management policies and associated decisions. More specifically, in order to increase carbon absorption, it is necessary to know the processes involved in the development of land surface layers and land use and to have solutions in hand to manage these processes. This can be achieved by assessing land-use development in Lithuania, with particular attention to the determinants of land use, understanding methodological principles for land-use development modelling. Wall-to-wall maps of land uses, developed at the compatible spatial and temporal resolutions using data in the Lithuanian National Forest Inventory, could help to improve both the evaluation of land-use status and the prediction of changes.

**Author Contributions:** Conceptualization, D.J. and G.M.; methodology, D.J. and G.M.; software, D.J. and G.M.; validation, G.M.; formal analysis, D.J. and G.M.; writing—original draft preparation, G.M., D.J. and V.K.; writing—review and editing, J.V. and V.A.; visualization, G.M. All authors have read and agreed to the published version of the manuscript.

**Funding:** This research received no external funding.

**Institutional Review Board Statement:** Not applicable.

**Informed Consent Statement:** Not applicable.

**Data Availability Statement:** Data available on request.

**Conflicts of Interest:** The authors declare no conflict of interest.

## Appendix A

**Table A1.** Factors used to model land-use development.

| Factor Name | Description | Date * | Source |
|---|---|---|---|
| Population density, 1989 | Population density in 1989, number of inhabitants/km$^2$ | 1989 | Population and housing census 1989 |
| Population density, 2011 | Population density in 2011, number of inhabitants/km$^2$ | 2011 | Population and housing census 2011 |
| Soil productivity grade | Average soil productivity score for agricultural land | | Dirv_DR10LT—spatial dataset of soil of the territory of the Republic of Lithuania at scale 1:10,000 |
| Land reclamation intensity | Drainage areas from the total area of the municipality, percentage | | Mel_DR10LT—spatial dataset of reclamation status and sodden soil of the territory of the Republic of Lithuania at scale 1:10,000 |
| Minimum altitude | Minimum altitude value within the borders of municipality | | Digital raster elevation model (cell size 100 m) in GDB200 GIS database—topographic map at scale 1:200,000. Elevation model was created using contour lines (interval between contours 20 m) and elevation points and applying Topo to Raster function of ArcGIS Desktop |
| Range of altitude | Range of altitude values within the borders of municipality | | |
| Mean altitude | The average altitude within the borders of municipality | | |
| Standard deviation of altitude | Standard deviation of altitude values within the borders of municipality | | |
| Mean slope | Average of terrain slope within the borders of municipality. Slope was calculated in degrees using Slope function of ArcGIS Desktop | | |
| Standard deviation of slope | Standard deviation of relief slope values within the borders of municipality. | | |
| Private land area, 2004 | Private land area in 2004 | 2004 | Agricultural census data, available from the Official Statistics Portal of Lithuania |
| Number of private owners, 2004 | Number of private owners in 2004 | 2004 | |
| Private land area per estate, 2004 | Average area of private land area per estate in 2004 | 2004 | |
| % of agricultural land in private land, 2004 | Proportion of agricultural land in private land area in 2004 | 2004 | |
| Private land area, 2008 | Private land area in 2008 | 2008 | |
| Number of private owners, 2008 | Number of private owners in 2008 | 2008 | |
| Private land area per estate, 2008 | Average area of private land area per estate in 2008 | 2008 | |
| % of agricultural land in private land, 2008 | Proportion of agricultural land in private land area in 2008 | 2008 | |
| Private land area, 2014 | Private land area in 2014 | 2014 | |
| Number of private owners, 2014 | Number of private owners in 2014 | 2014 | |
| Private land area per estate, 2014 | Average area of private land area per estate in 2014 | 2014 | |
| % of agricultural land in private land, 2014 | Proportion of agricultural land in private land area in 2014 | 2014 | |



**Table A1.** *Cont.*

| Factor Name | Description | Date * | Source |
|---|---|---|---|
| Grassland area per cattle-unit, 2008 | Area of permanent pasture for one animal unit in 2008 | 2008 | |
| Grassland area per cattle-unit, 2014 | Area of permanent pasture for one animal unit in 2014 | 2014 | |
| Forest, 1971 | Proportion of forest area in municipality in 1971 | 1971 | |
| Forest, 1990 | Proportion of forest area in municipality in 1990 | 1990 | |
| Forest, 2005 | Proportion of forest area in municipality in 2005 | 2005 | |
| Forest, 2015 | Proportion of forest area in municipality in 2015 | 2015 | |
| Producing land, 1971 | Proportion of producing land area in municipality in 1971 | 1971 | |
| Producing land, 1990 | Proportion of producing land area in municipality in 1990 | 1990 | Database of Lithuanian NFI |
| Producing land, 2005 | Proportion of producing land area in municipality in 2005 | 2015 | |
| Producing land, 2015 | Proportion of producing land area in municipality in 2015 | 2015 | |
| Grassland, 1971 | Proportion of grassland area in municipality in 1971 | 1971 | |
| Grassland, 1990 | Proportion of grassland area in municipality in 1990 | 1990 | |
| Grassland, 2005 | Proportion of grassland area in municipality in 2005 | 2005 | |
| Grassland, 2015 | Proportion of grassland area in municipality in 2015 | 2015 | |
| Population < 15-min drive to cities | Proportion of population residing within 15 min driving distance to cities | 2007 | Cartographic vector database of reference features according to the national specification KDB10LT-MIKRO (earlier version of current Georeference background cadastre (GRPK)), with all field and forest roads from Forest State Cadastre additionally included |
| Protection zones of roads | Area of protection zones around roads | | |
| Protection zones of railroads | Area of protection zones around railroads | | |
| Protection zones of electricity lines | Area of protection zones around electricity lines | | SŽNS_DR10LT—data base of limited land-use areas of the Republic of Lithuania at scale 1:10,000 |
| Protection zones of gas pipelines | Area of protection zones around gas pipelines | | |
| Protection zones of oil pipelines | Area of protection zones around oil pipelines | | |
| Graveyard protection zones | Area of graveyards and protection zones around them | | |

| Factor Name | Description | Date * | Source |
|---|---|---|---|
| Protection zones of water bodies | Area of protection zones around water bodies | | |
| Cultural heritage protection zones | Area of cultural heritage protection zones | | |
| Protected areas | Total area of protected areas | | |
| Area of abandoned land | Total area of abandoned agricultural land | | AŽ_DRLT—spatial dataset of neglected land of the territory of the Republic of Lithuania |
| Area of agricultural blocks, 2004 | Area of agricultural blocks in municipality in 2004 | 2004 | |
| Area of built-up blocks, 2004 | Area of built-up blocks in municipality in 2004 | 2004 | |
| Area of miscellaneous blocks, 2004 | Area of miscellaneous blocks in municipality in 2004 | 2004 | |
| Area of road infrastructure | Area of road blocks in municipality in 2004 | 2004 | |
| Length of streams, 2004 | Total length of streams in municipality in 2004 | 2004 | |
| Area of water bodies, 2004 | Area of blocks around the water bodies in municipality in 2004 | 2004 | |
| Area of agricultural blocks, 2008 | Area of agricultural blocks in municipality in 2008 | 2008 | |
| Area of built-up blocks, 2008 | Area of built-up blocks in municipality in 2008 | 2008 | Land parcel identification system (KZS_DR5LT) database and cartographic vector database of reference features according to the national specification KDB10LT-MIKRO or (for 2014) Georeference background cadastre (GRPK) |
| Area of miscellaneous blocks, 2008 | Area of miscellaneous blocks in municipality in 2008 | 2008 | |
| Area of road infrastructure, 2008 | Area of road blocks in municipality in 2008 | 2008 | |
| Length of streams, 2008 | Total length of streams in municipality in 2008 | 2008 | |
| Area of water bodies, 2008 | Area of blocks around the water bodies in municipality in 2008 | 2008 | |
| Area of agricultural blocks, 2014 | Area of agricultural blocks in municipality in 2014 | 2014 | |
| Area of built-up blocks, 2014 | Area of built-up blocks in municipality in 2014 | 2014 | |
| Area of miscellaneous blocks, 2014 | Area of miscellaneous blocks in municipality in 2014 | 2014 | |
| Area of road infrastructure, 2014 | Area of road blocks in municipality in 2014 | 2014 | |
| Length of streams, 2014 | Total length of streams in municipality in 2014 | 2014 | |
| Area of water bodies, 2014 | Area of blocks around the water bodies in municipality in 2014 | 2014 | |

* If no date is specified, the latest version of the relevant database was used.

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
