# Peer review of "Spatiotemporal Patterns of Land-Use Changes in Lithuania"

_land, doi:10.3390/land10060619_

Round 1
Reviewer 1 Report
The paper focuses on land-use change patterns in Lithuania by analyzing relevant data from 1971 to 2015. The manuscript is well-written and structured; it provides significant novelty and applies appropriate methodologies. The main strength of it can be grasped through the strong policy-oriented explanation of the revealed land use tendencies. The whole manuscript reflects the huge and comprehensive work that the authors did; therefore, it can be accepted after minor modifications; see my comments below:
- The Introduction section provides an adequate amount of background information regarding the selected topic and the Lithuanian scientific works. The only remark is related to the last paragraph: please 1) rephrase it and try to reduce its fragmented feature; 2) clearly specify the main aim(s) of the study;
- In Figure 1, some black shapes can be seen, marked as "not considered in the study". Can you provide an explanation of why these areas are excluded?
Author Response
Point 1: The paper focuses on land-use change patterns in Lithuania by analyzing relevant data from 1971 to 2015. The manuscript is well-written and structured; it provides significant novelty and applies appropriate methodologies. The main strength of it can be grasped through the strong policy-oriented explanation of the revealed land use tendencies. The whole manuscript reflects the huge and comprehensive work that the authors did; therefore, it can be accepted after minor modifications; see my comments below:
Response 1: Thanks for your evaluation
Point 2: The Introduction section provides an adequate amount of background information regarding the selected topic and the Lithuanian scientific works. The only remark is related to the last paragraph: please 1) rephrase it and try to reduce its fragmented feature; 2) clearly specify the main aim(s) of the study;
Response 2: The last paragraph of the introduction was modified following the comments of reviewer. More specifically, we start the paragraph introducing the main aim of the study: “The aim of current study is to map and explain the land-use changes in Lithuanian municipalities in the period since 1971”. We rephrased the remaining part of the paragraph aiming that next 3 sentences remaining could be read as short introduction of research tasks. We removed the text introducing “several objectives in this study”. Then, we combined 2 sentences into the following: “We mapped land use types that are considered the most significant in terms of carbon storage using land-use data originating from the Lithuanian NFI”. Then, we also added, that we evaluate and explain the factors behind the land-use change pattern.
Point 3: In Figure 1, some black shapes can be seen, marked as "not considered in the study". Can you provide an explanation of why these areas are excluded?
Response 3: We excluded 9 predominantly urban municipalities from the study, considering that their areas were too small to result in statistically comparable outputs with other municipalities using NFI data, the land-use structure and factors behind its change were considered to be different from the remaining municipalities. This is explained in the second paragraph of subchapter 2.2 Input data with a reference to Figure 1. Also, additional text was inserted in Figure 1 explaining that black shapes indicate preliminary urban municipalities.
Reviewer 2 Report
The manuscript is well written and suitable for publication in Land. Authors provided relevant information of data and methodology adopted, while results and discussion are well presented. However, there a few issues that need attention prior to its final acceptance.
Page 5, line 192: Please provide definition for SD
Caption of Figure 3 is not that clear
Consider providing more details regarding the models mentioned in Table 2.
Author Response
Point 1: The manuscript is well written and suitable for publication in Land. Authors provided relevant information of data and methodology adopted, while results and discussion are well presented. However, there a few issues that need attention prior to its final acceptance.
Response 1: Thanks for your evaluation. We tried to fix all the issues mentioned in your comment.
Point 2: Page 5, line 192: Please provide definition for SD
Response 2: Abbreviation SD was changed into “standard deviation”.
Point 3: Caption of Figure 3 is not that clear
Response 3: Not sure whether we correctly understood the comment. We adjusted the caption following the one used for Figure 4 (making them more compatible). I.e.: “Area proportions of land-use types in Lithuanian municipalities during different periods since 1971”. The remaining text in the caption is aimed to explain specific formatting or use of some symbols. The formulation of this text was also slightly adjusted (also in Figure 4).
Point 4: Consider providing more details regarding the models mentioned in Table 2.
Response 4: Coefficients of regression models included.
Reviewer 3 Report
Figure 1 needs to have a Legend that does not overlap with nearby map.
I see the tow different maps are shown with the combined Legend.
I would highly recommend to revise the map.
Overall I think the manuscript is well structured.
Author Response
Point 1: Figure 1 needs to have a Legend that does not overlap with nearby map.
I see the tow different maps are shown with the combined Legend.
I would highly recommend to revise the map.
Overall I think the manuscript is well structured.
Response 1: Figure 1 was adjusted in the following way – the legend which referred to both central and right portions of the figure was moved down and place below both smaller maps. Additionally, the legend was expanded following the comment of Reviewer 1.